# Photothermal Quotient Describes the Combined Effects of Heat and Shade Stresses on Canola Seed Productivity

Gonzalo M. Rivelli [1], Nora V. Gomez [2], Anita I. Mantese [2], Daniel J. Miralles [1], Leonor G. Abeledo [1] and Deborah P. Rondanini [1,*]

[1] IFEVA, CONICET, University of Buenos Aires, Av. San Martin 4453, Buenos Aires C1417DSE, Argentina
[2] School of Agriculture, University of Buenos Aires, Av. San Martin 4453, Buenos Aires C1417DSE, Argentina
[*] Correspondence: rondanin@agro.uba.ar

**Abstract:** There is evidence of the negative effects on canola seed yield caused by shading (SH) and high temperature stress (HT) separately, but the combined effect of both stresses has not been studied. This work aimed to (i) evaluate the effects of SH and HT stresses, alone and combined, on floral development, seed yield and quality, (ii) quantify the resulting effect (additive, synergistic, antagonistic) of combined stresses, and (iii) examine the utility of the photothermal quotient (PTQ, solar radiation/temperature ratio) to predict seed yield in stressed canola crops. Two field experiments were performed in Buenos Aires (Argentina) applying HT daytime temperature stress (25–30 °C from 1000 to 1500 h), SH (−80% irradiance), and SH + HT combined stresses, with C unstressed (20 °C and 100% irradiance) crops. Long and short duration SH and HT strongly affected floral development (fewer flowers and pods, with smaller ovules) and seed yield (reduction from −40 to −90% respect to C). Combined SH + HT exhibited detrimental synergistic effects on seed yield and oil concentration for long duration stresses, whereas antagonistic effects were mainly observed for short stresses. We conclude that the PTQ (cumulative from 100 to 500 °Cd after flowering) summarizes adequately the detrimental effects of combined post-flowering abiotic stresses on canola seed productivity.

**Keywords:** rapeseed; *Brassica napus*; solar radiation; high temperature; flowering; photothermal quotient; grain yield; seed weight; seed oil; fruiting efficiency





## 1. Introduction

Canola (*Brassica napus* L.) is the third global oilseed crop after soybean and oil palm and it is valued for its high seed oil content and excellent nutritional quality. In many regions around the world there is an increased interest in this crop, associated with more complex and intensified farming systems [1–4]. High variability in canola seed yield has been documented at both global [5–7] and local scales, mainly related to environmental effects and the genotype x environment interaction [8–12].

Most of the variability in canola seed yield is explained by the occurrence of adverse environmental factors during the critical stages of the crop cycle [13]. In this sense, non-optimal levels of solar radiation around flowering reduce seed yield. The detrimental effects of shading during grain filling on seed yield and quality have been observed in chamber [14] and in field experiments [15–21]. The duration of stress can cover the entire post-flowering period or just part of it. Kirkegaard et al. (2018) demonstrated, by applying brief shading treatments, that the critical period for grain yield definition in canola extended from 100 to 500 °Cd after the beginning of flowering, which is the period of greatest sensitivity to shading stress; while Dreccer et al. (2018) reported that the most sensitive period extended 200–400 °Cd after the beginning of flowering [20,22].

Floral development, pod setting, and seed viability are crucial processes for determining canola seed yield, and they are sensitive to high temperature [23]. In chamber experiments it was observed that seed yield is adversely affected by heat stress [24–28]. In

addition, the negative effects of high night temperature and maximum day temperature above 30 °C during grain filling on yield have been observed in field trials [22,29,30]. Other reports showed that the number of flowers is adversely affected by heat stress [24–27,31,32] and by shading [14] at the early flowering stage. Around 30% of the ovules are sterile, due to the absence of the embryo sac [33], and cultivars became almost entirely sterile when canola plants were grown in chambers at 27/17 °C and 32/26 °C, the phase most sensitive to heat stress being from late bud to early seed development [26,34]. The processes mainly affected by high temperature are: reduced pollen viability and ovule fertility, abortion of flowers, and ovary damage [24,31,33]. All these processes finally reduce the biomass allocation to reproductive organs, quantified by the harvest index [13,16]. In addition, fruiting efficiency (i.e., the number of seeds per gram of non-seed reproductive biomass) has proven to be a useful variable that reflects biomass partitioning within reproductive structures in cereals [35] and canola [36].

As indicated above, there are several reports of the negative effects on seed yield caused by shading and heat stress separately, but the combined effect of both stresses has not been studied in depth in canola. It is relevant to consider that in nature, adverse environmental factors rarely occur in isolation but frequently coincide in time. From the background in the literature, it is clear that there are not enough systematic field experiments involving variations in the intensity and duration of shading and heat stresses on canola. The occurrence of both stresses seems to be increasing, associated with the higher frequency of extreme temperature events [37,38] and the global dimming [39–41] predicted in most climate change scenarios. Moreover, the resulting combined effect of both stresses cannot be estimated from each individual stress, since they may cause additive, synergistic effects; or, one stress may prevail over the other, as was observed for other combined stresses on model oilseed plants [42–44] and canola [28]. Although the photothermal quotient, i.e., the solar radiation/mean temperature ratio, is a variable that synthesizes the two environmental factors that best predict seed yield in several temperate crops including canola [22], its utility under combined stress conditions is unknown.

This study aimed to (i) evaluate the effects of shading and heat stress, alone and combined, on floral development, fruiting efficiency, seed yield and quality, (ii) quantify the resulting effects (additive, synergistic, antagonistic) of combined stresses on the productivity of field-grown canola, and (iii) examine the utility of the photothermal quotient to predict seed yield in stressed crops. It is hypothesized that (i) seed yield is significantly reduced by both types of abiotic stresses, heat and shade, mainly lessening the reproductive capacity, (ii) their joint occurrence affects in a greater magnitude than those of each individual stress, and (iii) seed yield is positive and linearly associated with the photothermal quotient for all situations of heat stress, shading, and their combination.

## 2. Materials and Methods

### 2.1. Experimental Details

Two experiments were carried out under field conditions at the School of Agriculture, University of Buenos Aires (34°35′ S, 58°29′ W, 25 m altitude) on a silty clay loam classified as a Vertic Argiudoll (USDA Soil Taxonomy). The experimental site has a humid temperate climate, influenced by the extensive Rio de la Plata river, with an average annual temperature of 18.4 °C and 1200 mm of annual cumulative precipitation (average for 1991–2022). Over time, the weather has become rainier (with an increase of 47 mm per decade since 1960) and hotter (the mean temperature has increased 1 °C since 1960) with an increase in the frequency of heat waves (i.e., heat waves doubled between 2010 and 2018, reaching 16 heat wave events). The cultivar Hyola 61 (Advanta Seed Co., Buenos Aires, Argentina) was sown on 18 May 2011 (Exp. 1) and 8 May 2012 (Exp. 2) on plots of seven rows with 0.2 m row spacing and 2 m long at 60 plants $m^{-2}$. Plots were fertilized at the two-leaf stage with 100 kg N $ha^{-1}$, 20 kg P $ha^{-1}$, and 15 kg S $ha^{-1}$. Rainfall was complemented by drip irrigation to avoid water stress. Pests and diseases were chemically controlled, and weeds were mechanically controlled.

Mean daily temperature (°C) and daily global incident irradiance (MJ m$^{-2}$ d$^{-1}$) values were obtained from a National Weather Service station (Villa Ortuzar, Buenos Aires, Argentina) located 200 m from the experiments. Average mean temperature during pre- and post-flowering periods was 11.6 and 16.9 °C, respectively, for Exp. 1; whereas it was 12.5 and 16.5 °C for Exp. 2. Average incident solar radiation was 8.4 and 16.9 MJ m$^{-2}$ d$^{-1}$ for Exp. 1, and 7.4 and 14.0 MJ m$^{-2}$ d$^{-1}$ for Exp. 2, during pre- and post-flowering periods, respectively. Detailed dynamics of air temperature and incident global solar radiation for both experimental years are shown in Figure 1.

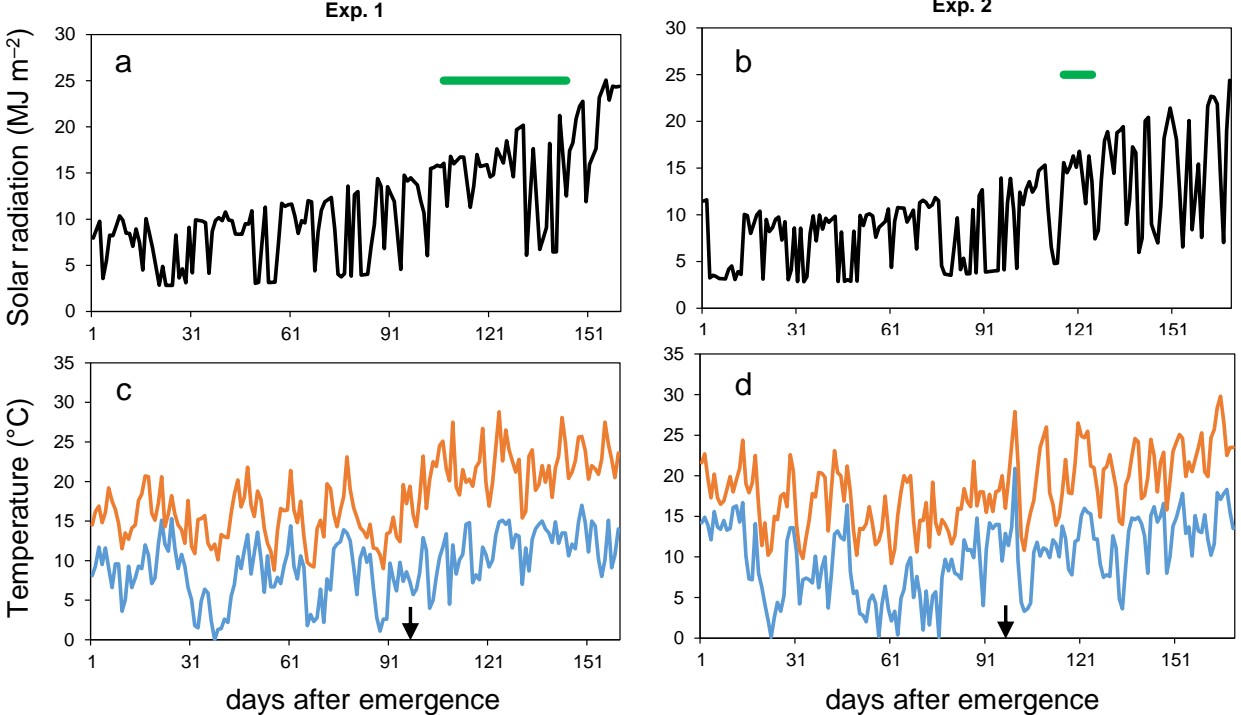

**Figure 1.** Dynamics of incident global solar radiation (**a**,**b**) and maximum and minimum daily air temperature (**c**,**d**) for Exp. 1 and 2. Black arrows indicate the beginning of flowering. Horizontal bars indicate the timing of the stress treatments.

### 2.2. Heat, Shading, and Combined Treatments

Treatments consisted of the factorial combination of high daytime temperature and shading, resulting in high temperature stress (HT), shading stress (SH), and combined stresses (SH + HT). A Control (C) without stress was also included. Timing and duration of treatments contrasted between experiments, with 37 days for Exp. 1 and 9 days for Exp. 2, respectively. They were applied from 8 to 45 days after flowering, DAF (Exp. 1), and from 17 to 26 DAF (Exp. 2). Expressed in thermal time, after flowering (temperature base = 0 °C) treatments were applied from 122 to 746 °Cd in Exp. 1, and from 244 to 401 °Cd in Exp. 2 (Figure 2). In both experiments, treatments overlapped the critical periods of 100–500 °Cd after flowering [20], and 200–400 °Cd after flowering [22].

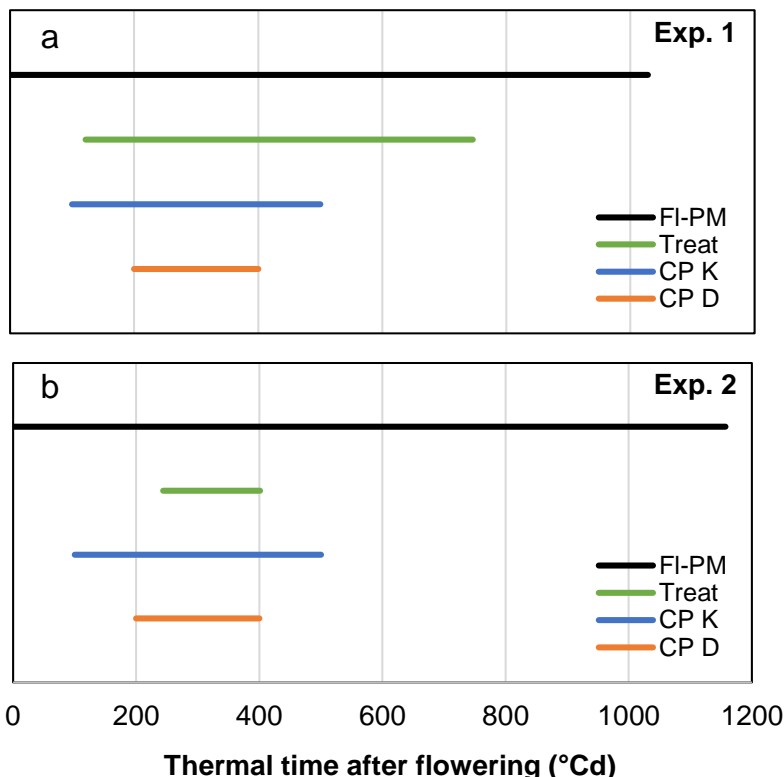

**Figure 2.** Thermal time from the beginning of flowering to physiological maturity (Fl-PM), timing of treatments (Treat) applied to canola in Exp. 1 (**a**) and 2 (**b**), and the critical periods (CP K) reported by Kirkegaard et al. (2018), and (CP D) by Dreccer et al. (2018) [20,22].

Portable chambers (2 × 1.4 × 1.7 m length, width, and height, respectively) with a steel framework were placed on plots (Figure S1). The height of the chamber ensured a distance of 0.5 m between the top of the crop canopy and the roof of the chamber (to avoid overheating the upper canopy). The HT treatment was applied covering the chambers with transparent polyethylene (200 μm thickness), equipped with an electric fan heater inside, connected to an automatic control unit (Cavadevices, Buenos Aires, Argentina) and monitored by temperature sensors (HOBO DTU10-003, Onset Corp., Bourne, MA, USA) placed at the height of main floral raceme into the canopy. The heaters in the chambers were set to increase the maximum daily temperature only during the five central hours of the day (from 1000 to 1500 h), up to a maximum temperature of 35 °C (a thermal threshold for oilseed crops according to [45]). During the heating period, the fan operated continuously and the plastic cover was opened for 5 min every hour in order to maintain normal levels of air moisture and gas concentration inside the chamber and to allow the entry of pollinators. The chambers were kept open during the rest of the day by rolling up the plastic sides. These actions aimed to avoid artifacts by manipulating the temperature [46,47].

The SH treatment was applied covering the chambers with black nets, reducing incident solar radiation by 80% in a similar way to [19]. The south face of the chamber remained without a net, to allow the visit of pollinators. The combined stress treatment (SH + HT) was applied covering the portable chambers with both black nets and plastic, with an electric heater inside, identical to that described for the HT treatment. The Control treatment (C) was applied covering only the roof of the chamber with plastic during the treatment period, in a similar way to [48]. Independent measures into the same type of growth chambers showed no significant changes among treatments in the $CO_2$ air concentration at the canopy height during the five central hours of the day, or when considering all the hours of the day (Rivelli GM, personal communication).

*2.3. Measurements*

Crop developmental stages were registered when 50% of the plants reached the stages of: crop emergence (16 and 23 May), beginning of flowering (23 August and 2 September), start of fruiting (3 and 9 September), and crop maturity (31 October and 1 November, for Exps. 1 and 2, respectively) according to the scale described by [49]. Physiological maturity (PM) was accurately determined by following the dynamics of seed dry weight of pods from the main floral raceme according to [50]. Durations of the phenological phases were expressed in both calendar (days) and thermal time units (base temperature = 0 °C).

For all treatments, incident global solar radiation at the top of the canopy and canopy intercepted solar radiation (ISR) were measured at noon on clear days twice a week during the whole crop cycle, using a 1-m long linear radiometer (Cava-Rad, Cavadevices, Buenos Aires, Argentina). The cumulative ISR for the whole crop cycle and for the post-flowering period (from first flowering to physiological maturity) were also calculated according to [51]. The reduction in incident solar radiation caused by the treatments (black nets and/or transparent polyethylene) was also measured. Air temperature (in both experiments) and relative humidity (in Exp. 2 only) were registered hourly by sensors placed at the height of the main floral raceme into the canopy (HOBO DTU10-003, Onset Corp., Bourne, MA, USA) linked to a datalogger. The photothermal quotient (PTQ) during the critical periods of 100 to 500 °Cd [20], and 200 to 400 °Cd [22] from the beginning of flowering, was calculated as the ratio between incident global solar radiation (MJ m$^{-2}$ d$^{-1}$) and daily mean temperature (°C), and was expressed as daily average value (MJ m$^{-2}$ d$^{-1}$ °C$^{-1}$) and as cumulated value (MJ m$^{-2}$ °C$^{-1}$). Vapour pressure deficit (VPD) was also calculated for treatments in Exp. 2 (from the relative humidity values measured hourly in the chambers), and the photothermal quotient adjusted by vapour pressure deficit (PTQ VPD) was calculated according to [22] which ponders the differential effect of day/night temperatures (75/25) on crop growth.

At first flowering, the leaf area index (LAI) was measured on all plants from 1-m central row of each plot, using a LI-3100C area meter (LI-COR Inc., Lincoln, NE, USA). Additionally, the red and far red ratio of the light (µmol m$^{-2}$ s$^{-1}$) was measured at soil level using a four-channel sensor SKR 1850A (Skye Instruments Ltd., Powys, UK), and the chlorophyll content of the upper petiolate leaves was measured with a SPAD-502Plus (Konica Minolta, Warrington, UK). These measurements served to establish the status of the canopy prior to the application of the stress treatments.

For floral development, three plants per plot were tagged and the dynamic of reproductive organs (flowers and pods) in the main raceme was followed by non-destructive observations until maturity, according to [26]. The morphology of young pods (<1 cm length) and older pods (>1 cm length) was examined. Reproductive organs were collected for all treatments at 10, 17 and 27 days (Exp. 1), or 10, 14 and 21 days (Exp. 2), after first flowering. The sampled material was immediately fixed in formalin/acetic acid/ethanol for 48 h, dehydrated in an ethanol/xylol series, and then infiltrated and embedded in paraffin and sectioned (10–12 µm thick) using a Minot-type rotary microtome. The sections were stained with safranin/fast green in ethanol, mounted in Canada balsam, and photographed with a Zeiss Axioplan optical microscope (Oberkochen, Germany) equipped with the Zeiss AxioCam ERc 5s software (Jena, Germany). The number and morphology of viable ovules were recorded [14].

At crop maturity, 0.9 m$^2$ from the three central rows of each plot (avoiding border plants) was harvested and above-ground biomass was dried in a forced-air oven at 70 °C for 72 h, weighed and threshed. The harvest index was calculated as the seed to above-ground biomass ratio. Seed yield (on a dry weight basis) was expressed on an area basis (g m$^{-2}$). Thousand seed weight was estimated from three 200-seed aliquots. The seed number per unit area was calculated as the seed yield divided by individual seed weight. Fruiting efficiency was calculated as the seed number produced by unit of non-seed reproductive biomass (also called chaff), according to [36]. Seed oil concentration (on a dry basis) was determined by Soxhlet extraction [36].

*2.4. Data Analysis*

For each experiment, treatments were arranged in a completely randomized design (CRD) with four replicates (Figure S1). The experimental unit was the individual plot and the significance of the differences between means of treatments were determined using ANOVA and Tukey's test at 5% level of significance. Linear regression analysis and Pearson's correlation analysis was also applied to the relationships between variables. Unless otherwise indicated, the mean $\pm$ 1 standard error are reported. Data transformation of percentage data (seed oil concentration) was applied to obtain homoscedasticity. Statistical packages were InfoStat (www.infostat.com.ar, accessed on 1 December 2022) and Graph Pad Prism (www.graphpad.com, accessed on 1 December 2022).

To qualify the effects of the combined versus the single stressors on the yield traits, calculations were made according to [28]. The effect weights of shading (SH), high temperature (HT), and shading + high temperature (SH + HT), compared to the unstressed Control (C) treatment, were calculated using the following formula:

$$Te = \left(Xt - \overline{X}c\right)/\overline{X}c \tag{1}$$

where Te is the treatment effect weight (its module), Xt is the trait "X" value for the treatment t, and $\overline{X}$ c is the corresponding mean value for the Control. The shading + high temperature effect weight obtained with the above formula (SH + HTe) was compared to the calculated shading + high temperature effect (SH + HTcalc) using the following formula:

$$HTcalc = \overline{SH}e + \overline{HT}e - \overline{SH}e \times \overline{HT}e \tag{2}$$

where $\overline{SH}$e and $\overline{HT}$e are the means of the shading and high temperature effect weights, respectively.

Comparing SH + HTe with SH + HTcalc, additive effects (both do not differ), synergistic effects (observed effect is greater than calculated), and antagonistic effects (observed effect is less than calculated) were identified respect to the individual effect of each stress [28].

## 3. Results

*3.1. Intensity and Duration of Stress Treatments*

Incident solar radiation and maximum daily temperature were effectively modified by the treatments (Figure 3). As expected, the SH treatment reduced radiation incident on the canopy by 81%, the plastic (in roof and laterals) of the chamber in HT reduced the incoming direct solar radiation by 27%, and the combined SH + HT darkened the canopy by 83% (Figure 3). Consequently, cumulative incident solar radiation during the treatment periods was strongly reduced with respect to C (Table 1). There was a close relationship between incident and intercepted solar radiation in both experiments, as the LAI at flowering ranged 4.9–5.3 (data not shown) and it maintained a high light capture efficiency (>0.95) throughout the treatment period. Additionally, the chlorophyll content of the leaves stayed high (39–43 SPAD units) and the red/far red ratio of light at soil level was very low, ranging from 0.07 to 0.14 (data not shown).

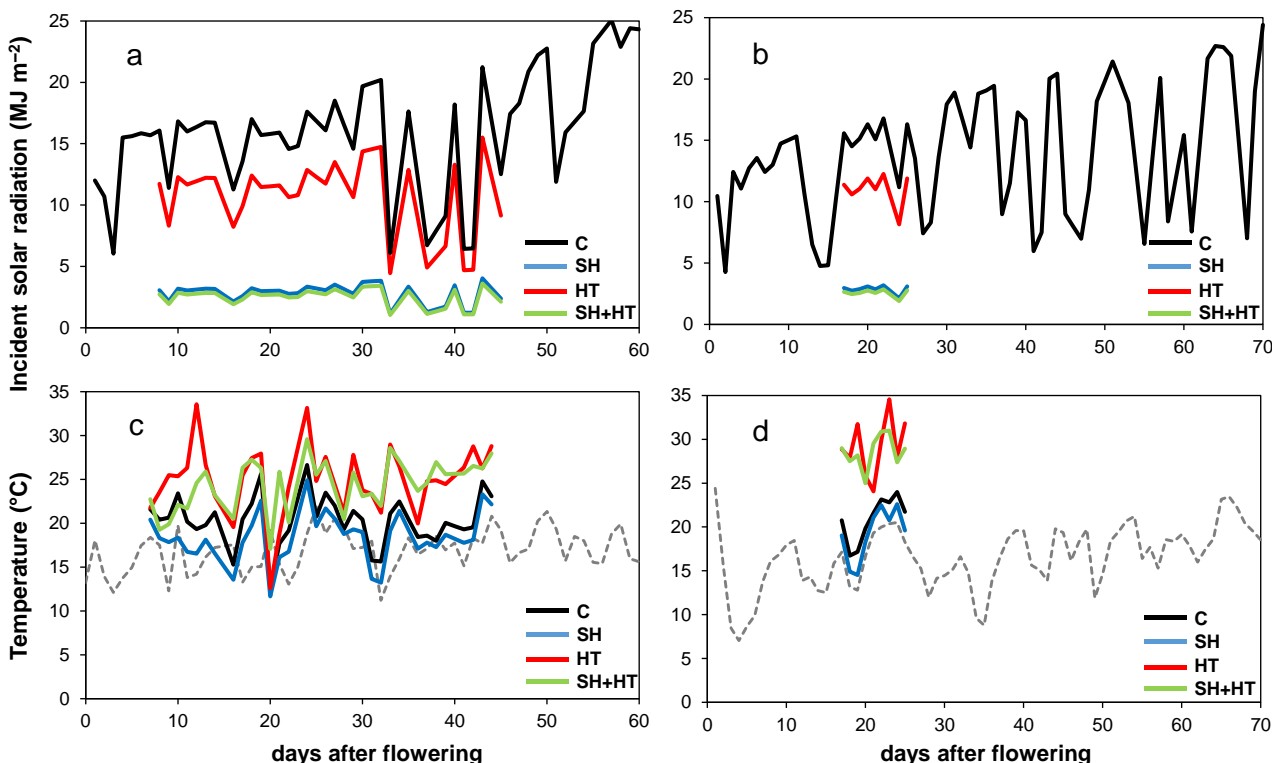

**Figure 3.** Dynamics of global solar radiation incident on the canopy (**a**,**b**) and temperature of the air inside the chambers during the hours of treatment (from 1000 to 1500 h) (lower panels) in Control (C), shading (SH), high temperature (HT), and combined stresses (SH + HT) in Exp. 1 (**a**,**c**) and 2 (**b**,**d**). Dashed line is mean daily temperature of the air outside the chambers.

**Table 1.** Cumulative incident solar radiation at the top of canopy within the chambers, minimum, maximum, and mean air temperatures in chambers during the treatment period (from 1000 to 1500 h), and average daily temperature considering the whole day (from 0 to 2400 h) in Control (C), shading (SH), high temperature (HT), and combined stresses (SH + HT) applied to canola in Exp. 1 and 2.

| Exp. | Treatment | Solar Radiation | Temperature from 1000 to 1500 h | | | Temperature from 0 to 2400 h |
|------|-----------|-----------------|---------------------------------|----|----|------------------------------|
| | | (MJ m$^{-2}$) | Minimum (°C) | Maximum (°C) | Mean (°C) | Mean (°C) |
| 1 | C | 556.0 ± 16 | 16.5 ± 4.1 | 24.5 ± 4.0 | 20.5 ± 4.0 | 17.5 ± 3.6 |
| | SH | 105.6 ± 8 | 16.2 ± 3.6 | 21.3 ± 3.6 | 18.8 ± 3.6 | 15.7 ± 3.5 |
| | HT | 405.9 ± 17 | 20.5 ± 5.6 | 30.9 ± 5.1 | 25.7 ± 5.3 | 20.3 ± 5.0 |
| | SH + HT | 94.5 ± 5 | 19.1 ± 4.4 | 30.1 ± 4.0 | 24.6 ± 4.3 | 19.8 ± 3.9 |
| 2 | C | 134.9 ± 4 | 20.0 ± 3.4 | 24.9 ± 3.1 | 20.9 ± 2.5 | 17.8 ± 2.2 |
| | SH | 25.6 ± 1 | 18.4 ± 2.6 | 21.3 ± 3.0 | 19.3 ± 2.9 | 15.8 ± 2.9 |
| | HT | 98.5 ± 4 | 21.7 ± 4.6 | 38.6 ± 3.4 | 29.2 ± 3.2 | 24.6 ± 3.3 |
| | SH + HT | 22.9 ± 1 | 20.6 ± 2.9 | 36.8 ± 2.9 | 28.6 ± 2.9 | 23.6 ± 2.4 |

Values are mean ± standard deviation.

Maximum temperatures > 30 °C were reached by heating in HT and SH + HT in Exp. 1, and especially in Exp. 2; so, HT in Exp. 2 was shorter but more intense than Exp. 1 (Figure 3). In both experiments, the temperature increase in the chambers during the noon hours (from 1000 to 1500 h) was variable among days (Figure 3), especially on sunny (greenhouse effect of the plastic roof) and windy days (turbulent mixing of the air into the chamber). The mean temperature from 1000 to 1500 h in the C treatment showed similar values in both experiments, with 20.5 and 20.9 °C (Table 1); and the black

nets in the SH treatment reduced mean temperature by 1.6 °C with respect to C. HT and SH + HT treatments increased the mean temperature from 1000 to 1500 h by 5.2 and 4.1 °C, respectively, in Exp. 1; whereas the temperature was increased by 8.3 and 7.7 °C, respectively, in Exp. 2. Because the chambers remained open the rest of the day, the increase in temperature over the whole day was less noticeable among treatments (see average daily temperature on the last column in Table 1).

The atmospheric demand (available for Exp. 2 only) was modified during the 5 h of treatment per day (see Supplementary Figure S2). The average VPD from 1000 to 1500 h was 1.2 kPa for C and was reduced to 0.4 kPa in the SH treatment (by reducing slightly the temperature and increasing the air humidity), while the HT had the opposite effect, increasing the VPD to 1.3 kPa (due to higher temperature and humidity). The SH + HT treatment had much more VPD (2.9 kPa) than C, associated with high temperature but air humidity similar to C. The rise of VPD was brought forward during the morning in the HT and SH + HT treatments, while it was delayed (approximately one hour) in SH (Figure S2). When calculating the atmospheric demand for the whole day, the differences between treatments were smoothed out, with average VPD from 0 to 2400 h of 0.7 kPa for C and HT, 0.5 kPa for SH and 1.1 kPa for SH + HT (Figure S2). Seed yield was not associated with VPD from 1000 to 1500 h ($p = 0.44$).

### 3.2. Effects of Shade and Heat Stresses on Floral Development

The dynamics of the appearance of flowers and pods on the main raceme was determined in both experiments. The maximum number of flowers observed in a particular day was reduced for individual and combined stresses, with respect to the Control (Figure 4). In addition, cumulative numbers of flowers and pods were strongly affected by treatments. The maximum cumulative number of pods in the main raceme was reduced by 28, 35, and 42% for SH, HT, and combined SH + HT, respectively, in the Exp. 1, with long durations of stresses; whereas the number of pods were reduced by 33, 41, and 50%, respectively, for the short and intense stress treatment of Exp. 2 (Figure 4). The final number of pods on the main raceme was reached around 50 days after flowering (Figure 4c,d) and crop maturity was around 60 and 69 days after flowering in Exp. 1 and 2, respectively.

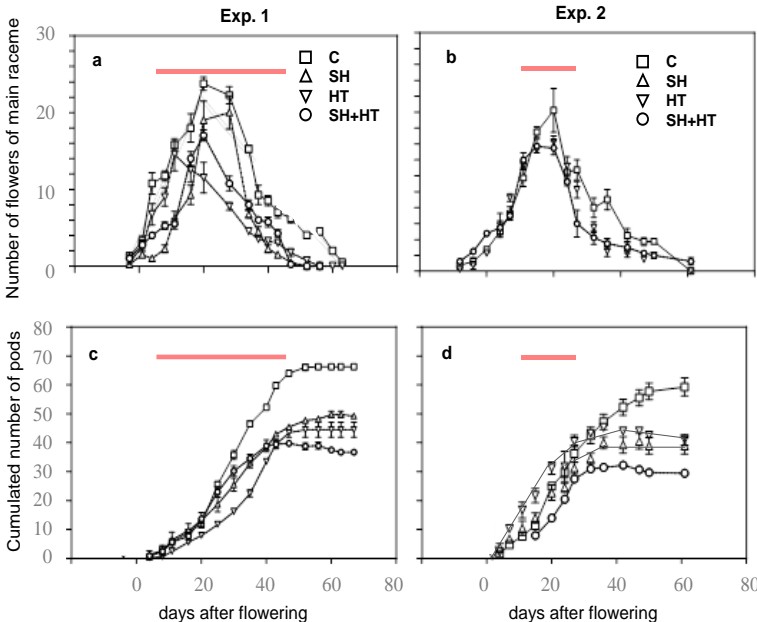

**Figure 4.** Dynamics of flowers in the main raceme (**a**,**b**) and cumulated number of pods (**c**,**d**) in Control (C), shading (SH), high temperature (HT), and combined stresses (SH + HT) in Exp. 1 and 2. Each point is the mean of 4 replicates ± 1 standard error. Horizontal bars indicate the timing of the stress treatments.

Viable ovules per pod (i.e., fertilized ovules that would be established as seeds number) from the main raceme were examined at different reproductive stages. The maximum number of ovules per pod was 16 and 18 in Exp. 1 and 2, respectively (data not shown). In young (<1 cm length) and older pods (>1 cm length), ovule size was reduced by the combined SH + HT with respect to the C, whereas the pericarp (pod wall) was slightly affected by treatments, as is shown in Figure 5 for Exp. 1.

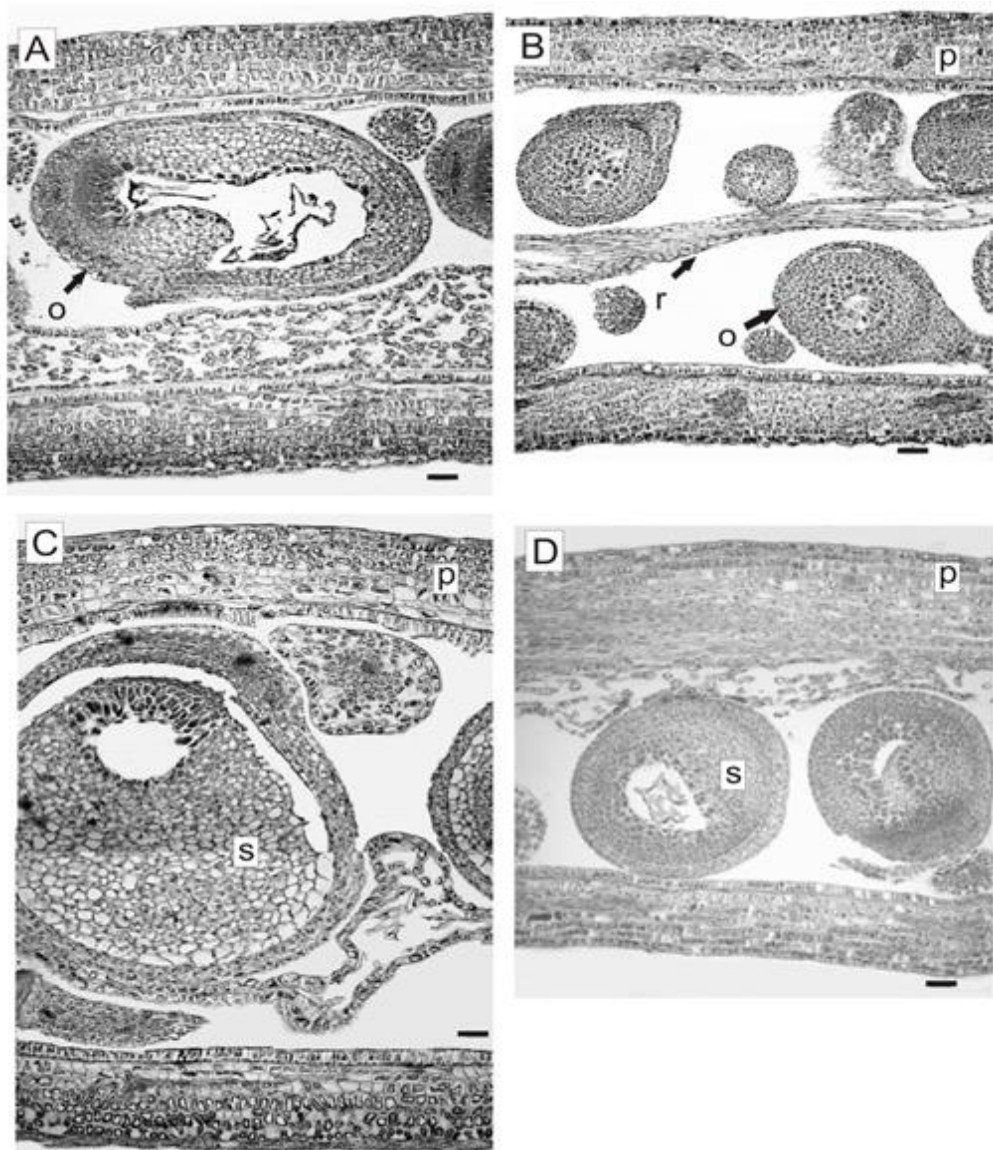

**Figure 5.** Micrographs (40×) of lengthwise sections of rapeseed pods from Exp. 1. (**A,B**) are young pods (<1 cm length), (**C,D**) are older pods (>1 cm length). Left panels are Control, right panels are combined shading and high temperature stresses (SH + HT). Abbreviations: (o) ovule; (p) pericarp; (r) replum; (s) seed. Scale bars = 100 µm.

### 3.3. Effects of Shade and Heat Stresses on Crop Productivity

Seed yield was significantly reduced by all stress treatments when compared to the C, in both experiments (Table 2). In Exp. 1 the lowest yield was in the SH + HT treatment, followed by both single stresses, without significant differences between them. In Exp. 2 single and combined treatments significantly reduced seed yield respect to C, without significant differences between SH, HT and SH + HT treatments (Table 2). Reduction of seed yield was closely correlated with seed number ($r = 0.98$; $p < 0.0001$), whereas seed

weight was mostly unaffected, except in the combined treatment (SH + HT) from both experiments. Seed oil concentration was significantly reduced by all the long duration stresses of Exp. 1, but not by the short duration stresses of Exp. 2 (Table 2).

**Table 2.** Seed yield, seed number and weight, seed oil concentration, above-ground biomass, harvest index, and fruiting efficiency in Control (C), shading (SH), high temperature (HT), and combined stresses (SH + HT) applied to canola in Exp. 1 and 2.

| Exp | Treatment | Seed Yield | Seed Number | Thousand Seed Weight | Seed Oil | Above-Ground Biomass | Harvest Index | Fruiting Efficiency |
|-----|-----------|------------|-------------|----------------------|----------|----------------------|---------------|---------------------|
| | | (g m$^{-2}$) | (10$^3$ m$^{-2}$) | (g) | (%) | (g m$^{-2}$) | | (seed g$^{-1}$) |
| 1 | C | 364.9 a | 109.8 a | 3.3 a | 42.9 a | 1165.7 a | 0.31 a | 227.9 a |
| | SH | 135.5 b | 39.6 c | 3.3 a | 37.4 b | 904.3 bc | 0.18 b | 119.1 bc |
| | HT | 213.9 b | 69.6 b | 3.1 a | 39.3 b | 958.0 ab | 0.23 b | 176.6 ab |
| | SH + HT | 39.5 c | 16.7 c | 2.4 b | 31.4 c | 476.4 c | 0.08 c | 69.2 c |
| | *p*-value | <0.0001 | <0.0001 | 0.0010 | <0.0001 | 0.0004 | <0.0001 | <0.0001 |
| 2 | C | 185.1 a | 55.2 a | 3.3 a | 39.2 a | 786.5 a | 0.24 a | 161.9 a |
| | SH | 113.7 b | 35.6 b | 3.2 a | 37.3 a | 563.3 a | 0.21 ab | 148.9 a |
| | HT | 94.2 b | 34.7 b | 2.7 ab | 36.6 a | 662.8 a | 0.15 b | 109.5 a |
| | SH + HT | 88.8 b | 39.6 ab | 2.3 b | 38.7 a | 525.6 a | 0.17 ab | 158.9 a |
| | *p*-value | 0.0024 | 0.0269 | 0.0014 | 0.1119 | 0.1822 | 0.0356 | 0.2562 |

For each experiment, means followed by different letters within a column indicate significant differences for Tukey's test. *p*-values are also shown.

Above-ground dry biomass at harvest was reduced by long duration stresses in Exp. 1, whereas no significant effects were observed for short-term stress in Exp. 2 (Table 2). Biomass partitioning was affected by stress treatments. Harvest index (the proportion of the above-ground biomass allocated to grain) was reduced by stresses, although the differences varied depending on the experiment. In Exp. 1 the highest penalization in harvest index was observed in the combined treatment. In Exp. 2, significant differences were observed between the C and HT treatments (Table 2). Significant correlation between seed yield and harvest index was observed for both experiments ($r = 0.87$; $p < 0.0001$). In turn, fruiting efficiency (the number of seeds per gram of non-seed reproductive biomass) was significantly affected in SH and SH + HT treatments from Exp. 1, but it was not affected by shorter treatments from Exp. 2 (Table 2). For the data set, harvest index and fruiting efficiency were significantly correlated ($r = 0.89$; $p < 0.0001$).

*3.4. Combined Shade and Heat Exhibit Synergistic Effects on Canola Productivity for Long-During Stresses*

The effects of the combined versus the single stressors on the yield traits were rated by comparing the observed effect weight of combined stresses (SH + HTe), with the calculated effect assuming additive effects (SH + HTcalc). For long-during stress in Exp. 1, synergistic effects of combined stresses were observed in all traits (Table 3), causing a greater detrimental effect than expected (SH + HTe > SH + HTcalc). By contrast, in short-duration stress from Exp. 2, antagonistic effects were mainly observed (Table 3), as the combined stresses caused less detrimental effect than expected (SH + HTe < SH + HTcalc). Thousand seed weight differed from the other traits, showing synergistic effects (Table 3).

**Table 3.** Effect weights of shading (SHe), high temperature (HTe), and combined stresses (SH + HTe) applied to canola in Exp. 1 and 2, compared to the calculated shading + high temperature effect (SH + HTcalc) assuming additive effects.

| Exp | Treatment | Seed Yield | Seed Number | Thousand Seed Weight | Seed Oil | Above-Ground Biomass | Harvest Index | Fruiting Efficiency |
|---|---|---|---|---|---|---|---|---|
| 1 | SHe | 0.63 | 0.64 | null | 0.13 | 0.40 | 0.42 | 0.48 |
| | HTe | 0.41 | 0.37 | 0.07 | 0.08 | 0.18 | 0.28 | 0.23 |
| | SH + HTe | 0.89 | 0.85 | 0.29 | 0.27 | 0.59 | 0.74 | 0.70 |
| | SH + HTcalc | 0.78 | 0.77 | 0.07 | 0.20 | 0.51 | 0.58 | 0.60 |
| | Result of combined stressors † | SI | SI | SI | SI | SI | SI | SI |
| 2 | SHe | 0.39 | 0.36 | 0.03 | 0.05 | 0.28 | 0.13 | 0.08 |
| | HTe | 0.49 | 0.37 | 0.18 | 0.07 | 0.16 | 0.38 | 0.38 |
| | SH + HTe | 0.52 | 0.28 | 0.32 | 0.01 | 0.33 | 0.29 | 0.02 |
| | SH + HTcalc | 0.69 | 0.59 | 0.20 | 0.11 | 0.40 | 0.46 | 0.37 |
| | Result of combined stressors † | AN | AN | SI | AN | AN | AN | AN |

† AD: additive, AN: antagonistic, SI: synergistic effect.

### 3.5. Photothermal Quotient Adequately Captures the Effects of Stresses on Seed Yield

Seed yield differences between treatments and experiments were adequately captured by the summary variable PTQ (Figure 6). All seed yield adjustments were highly significant ($p < 0.0001$), both for the PTQ in the critical period from 100 to 500 °Cd (CP K) and from 200 to 400 °Cd after flowering (CP D), as well as for the cumulated calculation and daily average, with the cumulated PTQ calculated for the period from 100 to 500 °Cd after flowering showing the best fit (Figure 6a). The cumulated PTQ from 100 to 500 °Cd after flowering also showed a high correlation with seed number ($r = 0.87$; $p < 0.0001$). When the PTQ was adjusted by the VPD (data from Exp. 2 only), no significant relationship was observed with seed yield (data not shown).

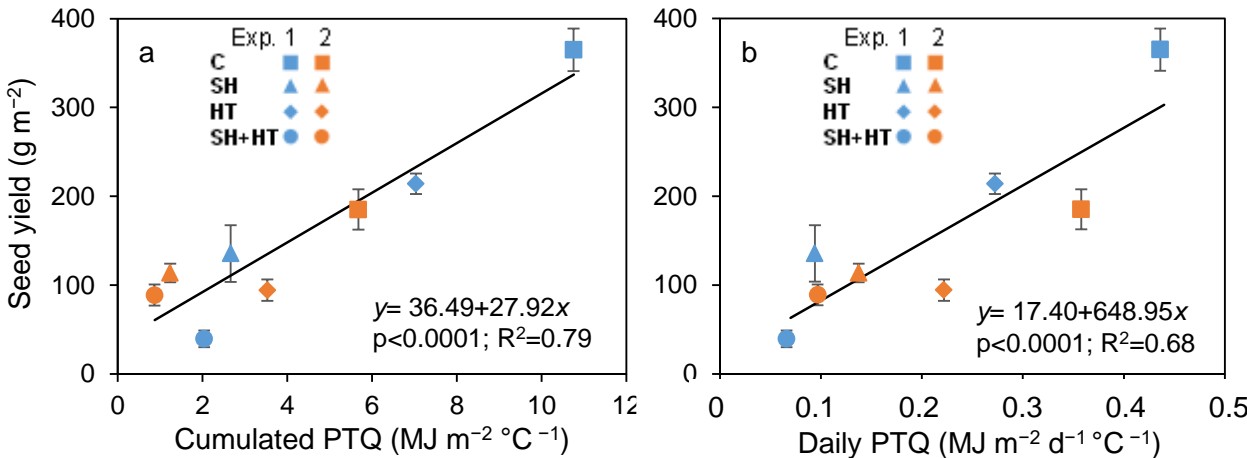

**Figure 6.** Relationship between seed yield and photothermal quotient (PTQ) cumulated during the critical period from 100 to 500 °Cd from the beginning of flowering (**a**) and average daily PTQ during 200 to 400 °Cd from the beginning of flowering (**b**) in Control (C), shading (SH), high temperature (HT), and combined stresses (SH + HT) applied to canola in Exp. 1 and 2. Each point is the mean of 4 replicates ± 1 standard error. Linear functions fitted to data are shown.

## 4. Discussion

Studies on the effect of abiotic stresses on crop productivity began several decades ago, evolving from trials in pots with individual plants affected by a single stress, to trials in field plots affected by multiple stressors. Although more complex and variable, field trials

with combined stresses are more similar to what farmers experience in actual cropping systems, allowing analysis of the mechanisms behind the response under realistic crop growing conditions [20,44,51].

Most of the studies of combined stresses on canola deal with drought and heat [28,52], which frequently occur during post-flowering in growing areas with a Mediterranean climate [13,22]. Studies on the combination of low solar radiation and high temperature are very scarce in canola [53], despite there being frequent adversities in several temperate climate cultivation areas, as in Southern South America [41], for which the present work intends to be a reference study.

As expected, high daytime temperature had strong effects on floral development, seed number and crop productivity, despite heating only 5 h a day. Such results are in line with recent findings [28,29] pointing out the high sensitivity of canola to heat stress and emphasizing the need to select heat-tolerant genotypes quickly [54]. Moreover, the detrimental effects of shading on seed yield (around −40%) agree with recent works that applied similar solar dimming (75–85%) over short times [19–21]. Thus, the observed results confirm the importance of the early post-flowering period, being that period very sensitive to abiotic stresses [16,20–22,29], for the survival of reproductive structures. In addition, our work describes the dynamics of flower and pod appearance under abiotic stress conditions, contributing to the advancement of knowledge about canola floral biology. Although pollen viability was not analyzed in the present work, there is a strong body of theory highlighting the direct detrimental effects of high temperature on pollen viability [24,29,34], the biochemical and molecular mechanisms of which are being elucidated [31,32]. On the other hand, studies of the effects of shading [14] and of the combined heat and shading stresses on the development of the male gametophytes and the viability of pollen are very scarce; further studies are required. Interesting effects of the long duration stresses were observed on total dry biomass, harvest index (biomass allocation to seed), and fruiting efficiency (reproductive biomass allocation to seed); this being the first time that changes in the reproductive partition in stressed canola have been documented.

Our results show that the combination of long-term shading and heat stress in the post-flowering period does not generate additive effects on seed productivity, but that the individual detrimental effects are exacerbated, generating responses of a synergistic nature. Considering that prolonged shading reduces the synthesis of photo assimilates (from leaves and pod wall), and that high daytime temperatures have direct effects on floral development [26–32] and multiple effects on crop growth and development (such as hastening senescence, reducing seed growth [13,45]), such synergy may be due to multiple simultaneously affected traits or to nullified or diminished stress mitigation processes. For example, relative to the unstressed condition, long-term SH and HT reduced seed number but maintained seed weight, while SH + HT reduced both seed number and weight, affecting both traits. Similarly, synergistic effects of the combined stresses were observed both in the amount of biomass and its allocation to reproductive structures.

By contrast, for short-term stress the combined effects were antagonistic in nature. This response was surprising, since the effect of the combined stress was less than expected, assuming additive effects. The antagonistic response may be related to the dominance of one stress over the other (so that when combining them, one does not contribute anything [28,44]) or to the occurrence of mitigation processes (which are not expressed under a single stress) such as the maintenance of the green area of pods, the remobilization and transport of assimilates, and the activation of synthesis enzymes in the seed, among others. In this way, a reduction in the source-sink ratio during canola seed filling induced the expression of genes involved in sucrose transport, seed weight, and stress responses [55].

Interestingly, thousand seed weight was affected little to none by any single stress, whereas the combined stresses had synergistic effects, regardless of stress duration. The sensitivity of seed weight affected by combined stresses may be associated with the direct effects of high daytime temperature on the rate and duration of seed growth (documented in sunflowers by [45]) exacerbated by a small potential seed size (associated with

poor maternal tissue growth of the pod and reduced seed volume, indicated in cereals by [56]) that could nullify the relative improvement in the source-sink ratio (caused by the drastic reduction in the number of seeds). Another possibility for synergistic effects on seed weight is the blocking of post-stress recovery processes. Direct evidence of these mechanisms is necessary in canola, considering the high reproductive plasticity of the species [19,36,55,57,58].

From the systematic knowledge of the effects of abiotic stresses on crop growth and seed development, it is possible to model the response to incorporate it into simulation models, expanding its usefulness to a range of genotypes and environmental conditions [59–61]. Environmental variables capable of summarizing stress intensity and duration are extremely useful for crop physiologists and modelers. Examples of such summary variables are the heat load and the PTQ. Heat load (calculated on daily or hourly basis above a threshold temperature of 30 °C) summarizes the effects of intensity and duration of heat stress, and is widely used in several crops [45,62–65]. The PTQ merges the positive effect of solar radiation on crop growth and the negative effect of temperature on the duration of the critical stages for the definition of seed yield [46], and has been shown to be associated with canola seed yield in rainfed Australian environments [22]. Our work expands the utility of the PTQ (cumulative from 100 to 500 °Cd after flowering) to capture the effects of shading and heat stress, individual and combined, for both short and long duration stress in canola seed yield. This is the first time in which the PTQ has been used for combined stresses, working properly in conditions of thermal stress, well above the optimal cardinal temperature of canola development [59]. In addition, the PTQ adequately captured the effect of the year on the seed yield of the unstressed controls, pointing out the importance of the variability of temperature and cloudiness recorded in the temperate environments of southern South America [41] underlying the instability of seed yields typical of this area [7,12]. Further studies should validate the association between the PTQ (cumulative from 100 to 500 °Cd after flowering) and seed development [66] and seed yield in a broader range of canola genotypes and stressful environmental conditions.

## 5. Conclusions

Shading and heat stress (for only 5 h a day) strongly affected floral development (fewer flowers and pods, with smaller ovules) and seed yield. Combined stresses exhibited detrimental synergistic effects on seed yield and oil concentration for long duration stresses, whereas antagonistic effects were mainly observed for intense and brief stresses. The photothermal quotient (cumulative from 100 to 500 °Cd after flowering) summarizes adequately the detrimental effects of combined post-flowering abiotic stresses on canola seed productivity, expanding its utility for crop physiologists and modelers.

**Supplementary Materials:** The following supporting information can be downloaded at: https://www.mdpi.com/article/10.3390/seeds2010012/s1, Figure S1: General view of the chambers placed on individual plots in Control (C), shading (SH), high temperature (HT), and combined stresses (SH + HT) in Exp. 1 (a) and Exp. 2 (b); and Figure S2: Dynamics of vapour pressure deficit (VPD) inside the chambers during treatments in Control (C), shading (SH), high temperature (HT), and combined stresses (SH + HT) in Exp. 2.

**Author Contributions:** Conceptualization, D.P.R., L.G.A. and D.J.M.; methodology, N.V.G., A.I.M. and D.P.R.; formal analysis and investigation, G.M.R., D.P.R. and D.J.M.; writing—original draft preparation, G.M.R. and D.P.R.; writing—review and editing, L.G.A. and D.J.M.; project administration, D.P.R.; funding acquisition, L.G.A. and D.J.M. All authors have read and agreed to the published version of the manuscript.

**Funding:** This research was funded by National Agency of Scientific and Technological Promotion of Argentina, grant number PICT 1294, and G.M.R. held scholarship from this Agency during this study.

**Institutional Review Board Statement:** Not applicable.

**Informed Consent Statement:** Not applicable.

**Data Availability Statement:** The data presented in this study are available on request from the corresponding author.

**Acknowledgments:** The authors thank A. Otero, L. Pedace, M. Rodriguez, and S. Enciso for technical assistance, and A.J. Hall for helpful comments on an early version of the manuscript.

**Conflicts of Interest:** The authors declare no conflict of interest. The funder had no role in the design of the study; in the collection, analyses, or interpretation of data; in the writing of the manuscript; or in the decision to publish the results.

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
