# Peer review of "Photothermal Quotient Describes the Combined Effects of Heat and Shade Stresses on Canola Seed Productivity"

_2674-1024, doi:10.3390/seeds2010012_

Round 1

Reviewer 1 Report

Introduction:

Line 76-80: this paragraph talks about the broad research question. Add a paragraph with specific prediction and hypothesis.

Methods:

Include annual temp and annual precipitation of the experiment site.

Results:

The distribution plot (e.g. histogram) for the different treatment might be useful to predict pattern of skewness.

For fig 4c,d – what happened after day 80. Is this data incomplete?

Discussion:

This sections needs to be improved by citing other similar discoveries and how is this relevant to the field. 

Author Response

Reviewer 1

Line 76-80: this paragraph talks about the broad research question. Add a paragraph with specific prediction and hypothesis.

Response: In accordance with the reviewer's comment, the following paragraph was added (in blue in L 80-84 in the new manuscript):

It is hypothesized that (i) seed yield is significantly reduced by both types of abiotic stresses, mainly lessening the reproductive capacity, (ii) that their joint occurrence affects in a greater magnitude than those of each individual stress, and (iii) that seed yield is positive and linearly associated with the photothermal quotient for all situations of heat stress, shading and their combination.

Methods:

Include annual temp and annual precipitation of the experiment site.

 Response: More context about the climate was added, and stress frequency in the experimental site was also included (as suggested by the reviewer #2). The following phrase was added (in blue in L 89-95 in the new manuscript):

The experimental site has a humid temperate climate, influenced by the extensive Rio de la Plata river, with an average annual temperature of 18.4 ºC and 1200 mm of annual cumulative precipitation (average for 1991-2022). Over time, the weather has become rainier (with an increase of 47 mm per decade since 1960) and hotter (the mean temperature increased 1ºC since 1960) with an increase in the frequency of heat waves (i.e. heat waves were doubled between 2010 and 2018, reaching 16 heat wave events).

Results:

The distribution plot (e.g. histogram) for the different treatment might be useful to predict pattern of skewness.

Response: No skewness was observed for the response variables (only the number of grains showed a slight positive asymmetry), and the assumptions of normality and homoscedasticity were satisfied for ANOVA. Normality was tested by the Shapiro and Wilk’s test, with p-values >0.01 for all variables.  

For fig 4 c,d – what happened after day 80. Is this data incomplete?

Response: The data in Fig. 4 c and d are complete, as they covered the whole period from the beginning of flowering to crop maturity. From 40 days after flowering, the opening of flowers on the main raceme was drastically reduced, and from 60 days after flowering no more flowers appeared (Fig. 4c). In addition, 50 days after flowering, the final number of pods on the main raceme was reached (Fig 4d). Crop maturity was reached at 60 days after flowering in Exp 1 and 69 days in Exp 2 (as it arises from the flowering and maturity dates reported in M&M), which is into the range (from 40 to 70 days) usually observed for post-flowering duration in spring canola genotypes in southern South America (10.1016/j.fcr.2017.05.021; 10.1016/j.fcr.2018.11.002).

To clarify the paragraph, the following phrase was added in L 301-303 in the new manuscript: The final number of pods on the main raceme was reached around 50 days after flowering (Fig. 4 c, d) and crop maturity was around 60 and 69 days after flowering in Exp 1 and 2, respectively.  

Discussion:

This section needs to be improved by citing other similar discoveries and how is this relevant to the field. 

Response: We agree with the reviewer’s comment. The Discussion was improved by adding the following paragraph (in blue in L 401-414 in the new manuscript):

Thus, the observed results confirm the importance of the early post-flowering period for the survival of reproductive structures, being that period very sensitive to abiotic stresses [16, 20-22, 29]. In addition, our work describes the dynamics of flower and pod appearance under abiotic stress conditions, contributing to the advancement of knowledge about canola floral biology. Although pollen viability was not analyzed in the present work, there is a strong body of theory highlighting the direct detrimental effects of high temperature on pollen viability [24, 29, 34] whose biochemical and molecular mechanisms are being elucidated [31, 32]. On the other hand, the study of the effects of shading [14] and for the combined heat and shading stresses on the development of the male gametophytes and the viability of the pollen are very scarce and require further study. Interesting effects of long duration stresses were observed on total dry biomass, harvest index (biomass allocation to seed) and fruiting efficiency (reproductive biomass allocation to seed), being the first time that changes in the reproductive partition for stressed canola crops have been documented.

Reviewer 2 Report

The manuscript is well written and includes a comprehensive data set. The research is interesting and focuses on the effects of heat stress and shading on reproductive traits in canola, an important oilseed crop.

I have only a few comments that may be of interest to readers:

- Since heat stress is associated with intense solar radiation, it would be useful to give more information on the geographical areas and climatic conditions where these two types of stress occur simultaneously. 

-  The experiments were repeated in two consecutive years and the design was complex. A picture of the experimental set-up would be useful to get a better idea.

Viable ovules per pod were analysed, but what about the male gametophyte? Did the authors analyse pollen viability or anther development under stress conditions? If not, is there any information in the existing literature on these aspects? Pollen viability is extremely susceptible to thermal shock, but to my knowledge there is little information on the effects of shading.

Author Response

Reviewer 2

The manuscript is well written and includes a comprehensive data set. The research is interesting and focuses on the effects of heat stress and shading on reproductive traits in canola, an important oilseed crop.

I have only a few comments that may be of interest to readers:

- Since heat stress is associated with intense solar radiation, it would be useful to give more information on the geographical areas and climatic conditions where these two types of stress occur simultaneously. 

Response: In accordance with the reviewer (and the reviewer #1 also) climatic data was expanded and the following phrase was added (in blue in L 89-95 in the new manuscript): The experimental site has a humid temperate climate, influenced by the extensive Rio de la Plata river, with an average annual temperature of 18.4 ºC and 1200 mm of annual cumulative precipitation (average for 1991-2022). Over time, the weather has become rainier (with an increase of 47 mm per decade since 1960) and hotter (the mean temperature increased 1ºC since 1960) with an increase in the frequency of heat waves (i.e. heat waves were doubled between 2010 and 2018, reaching 16 heat wave events).

-  The experiments were repeated in two consecutive years and the design was complex. A picture of the experimental set-up would be useful to get a better idea.

Response: For each year, the experimental design was a completely randomized design, with 16 individual plots to which the treatment was randomly assigned (as explained in section 2.4). We added photographs on the chambers fitted to individual plots, in order to improve the experimental set-up. In the text, it is mentioned as supplementary figure S1 (L 133 and L 215 in the new manuscript). Also, the former Figure S1 was renamed as Fig. S2 throughout the text.

- Viable ovules per pod were analysed, but what about the male gametophyte? Did the authors analyse pollen viability or anther development under stress conditions? If not, is there any information in the existing literature on these aspects? Pollen viability is extremely susceptible to thermal shock, but to my knowledge there is little information on the effects of shading.

Response: We agree with the reviewer’s comment. Unfortunately, we did not analyze pollen viability or anther development under stress conditions, due to lack of time to apply the correct techniques properly. Nevertheless, as it is mentioned in the Introduction, there is a solid body of theory on the direct detrimental effects of high temperature on pollen viability and ovule fertilization from several decades ago (see citations 24 and 34) to recent advances in modern canola genotypes and novel techniques to understand the biochemical and molecular responses to heat stress (see citations 31 and 32). There are less references on the effects of shade or light exclusion on pollen viability or floral fertility (we found only citation 14) although its final effect can be partially inferred by examining the final number of seeds achieved (which is reported in several works that modify the source-sink ratio). In the manuscript we added the following phrase (in L 405-414 in the new manuscript): Although pollen viability was not analyzed in the present work, there is a strong body of theory highlighting the direct detrimental effects of high temperature on pollen viability [24, 29, 34] whose biochemical and molecular mechanisms are being elucidated [31, 32]. On the other hand, the study of the effects of shading [14] and for the combined heat and shading stresses on the development of the male gametophytes and the viability of the pollen are very scarce and require further study. Interesting effects of long duration stresses were observed on total dry biomass, harvest index (biomass allocation to seed) and fruiting efficiency (reproductive biomass allocation to seed), being the first time that changes in the reproductive partition for stressed canola crops have been documented.  

Reviewer 3 Report

Manuscript entitled “Photothermal quotient describes the combined effects of heat and shade stresses on canola seed productivity” is well written. However, few major corrections needed to incorporate before publishing in reputed journal.

1.      In figure 5. Picture quality is poor. Can authors submit improved quality?

2.      Why authors have not included physiological performance of plants?

3.      Shade and heat also affecting dry biomass?

Author Response

Reviewer 3

Manuscript entitled “Photothermal quotient describes the combined effects of heat and shade stresses on canola seed productivity” is well written. However, few major corrections needed to incorporate before publishing in reputed journal.

  1. In figure 5. Picture quality is poor. Can authors submit improved quality?

Response: We are submitting the original figure (prior to be pasted in the Word template) as an attempt to improve its quality.

  1. Why authors have not included physiological performance of plants?

Response: We were mainly interested on physiological responses at crop level (for the entire canopy) instead of focusing on physiological responses at plant level. In order to give accuracy to the work and given the level of organization that was under study, the experiments were carried out in plots at field. The complete effects of stresses on carbon assimilation were analyzed through the above-ground biomass at harvest and the biomass partitioning to reproductive organs (fruiting efficiency and harvest index), which are accepted physiological variables at the crop level. Such responses were emphasized by adding the following phrases in the Results and Discussion sections: L 332-333: Above-ground dry biomass at harvest was reduced by long duration stresses in Exp. 1 whereas no significant effects were observed for short-term stress in Exp. 2 (Table 2). L411-414: Interesting effects of long duration stresses were observed on total dry biomass, harvest index (biomass allocation to seed) and fruiting efficiency (reproductive biomass allocation to seed), being the first time that changes in the reproductive partition in stressed canola have been documented.

Unlike the study of abiotic stresses on leaf photosynthesis in other crops that maintain the leaf area during mostly of the post-flowering period, in canola the leaf area decreases rapidly after flowering and photosynthesis is quickly replaced by the green area of ​​pods. Thus, it could be of interest to perform physiological measurements of carbon uptake in pods under stress, but unfortunately we don’t have the proper gas exchange chamber (which is larger than that available for Arabidopsis siliques). On the other hand, exploratory measurements of leaf fluorescence were made before and after the application of the stress treatments, without conclusive results (low fluorescence for shade, similar fluorescence for control and heat, compatible with the fact that leaves were located at the bottom of canopy, in a zone that was already shaded and in advanced state of leaf senescence) for which reason they were not included in the manuscript.

  1. Shade and heat also affecting dry biomass?

Response: Yes, shade and heat affected total above-ground dry biomass at maturity for long duration stresses in Exp. 1 (see Table 2, column ‘Above-ground biomass’) whereas no significant effects were observed for short-term stress in Exp. 2. In the manuscript we added the following phrase (in L 332-333 in the new manuscript): Above-ground dry biomass at harvest was reduced by long duration stresses in Exp. 1 whereas no significant effects were observed for short-term stress in Exp. 2 (Table 2).

Round 2

Reviewer 3 Report

Authors have incorporated all the suggestions. I recommend manuscript to publish in current version.